# Shared Religious Education through Christian–Islamic Team Teaching

Agnes Gmoser [1], Michael Kramer [2], Mevlida Mešanović [1], Wolfgang Weirer [1,*], Eva Wenig [1] and Şenol Yağdı [2]

[1] Department for Catechetics and Religious Education, University of Graz, 8010 Graz, Austria; agnes.gmoser@uni-graz.at (A.G.); mevlida.mesanovic@uni-graz.at (M.M.); eva.wenig@uni-graz.at (E.W.)
[2] Department of Islamic-Theological Studies, University of Vienna, 1010 Vienna, Austria; michael.kramer@univie.ac.at (M.K.); senol.yagdi@univie.ac.at (Ş.Y.)
* Correspondence: wolfgang.weirer@uni-graz.at; Tel.: +43-316-380-6232

**Abstract:** The article, which is written by an interreligious team, provides comprehensive insights into the conception, implementation and accompanying research of a project on Christian–Islamic religious education in team teaching. The aim of the project is to expand the denominational religious education lessons anchored in Austrian schools through religious-cooperative units taught jointly by a Christian and an Islamic teacher. The analysis of the teaching units is carried out in the format of design-based research and thus encompasses numerous aspects of interreligious educational processes, which are examined in this article. Firstly, the design of the project is described and the legal framework associated with it is explained in the context of the Austrian school system. Subsequently, learning requirements on the part of Christian and Muslim pupils are presented, with a particular focus on their preconceptions and attitudes towards religion in general as well as other religions. Special attention is paid to the specific framework conditions of Islamic religious education teachers, which differ in many aspects from those of Catholic religious education teachers. Furthermore, interreligious competences they consider necessary are described. Specific insights into the teaching units and the complementary research provide information about the opportunities and challenges of interreligious education in team teaching by two teachers. After this focus on the teachers, an outline of the students' perspectives on the teaching units completes the presentation of the research results. In the concluding summary, the local theories developed from the overall project are presented and discussed.

**Keywords:** interreligious learning; interfaith dialogue; team teaching; religious education; Christians; Muslims; design-based research

## 1. Introduction

Over the last decade, diversity has significantly increased across the student body in schools in Austria. Several migration movements since 2015 have contributed to a general rise in religious and ethnic diversity, and this is especially evident in the context of education. These changes present challenges for the education system as a whole, as well as religious education in particular.

In many schools, denominational religious education alone cannot meet these emerging organisational and conceptional demands. In response, religious pedagogy has been exploring cross-denominational and interreligious collaborative teaching models for several years (Krobath and Taschl-Erber 2023; Woppowa and Käbisch 2023; Schambeck 2022). At the same time, "religious education" has been the subject of fundamental academic reflection (Unser 2021, 2018; Meyer 2019; Meyer and Tautz 2015) over the last two decades, while various models of interreligious celebrations, projects and collaborative teaching are being trialled in various contexts. However, theory and practice have largely developed in



parallel, with little intersection. Academic concepts remained theoretical and untrialled in classrooms, while religion teachers created their own solutions for interreligious learning and teaching to meet structural and didactic challenges.

In Austria, religious education is a compulsory subject across all school types and grades and is delivered according to denomination. However, students can opt out, and those who do not belong to a specific denomination can choose to opt in. Responsible for teaching content and the delivery of religion classes are the respective, officially recognised church and religious communities (currently 16, with 15 offering denominational religious education). This legal framework is unique among other central European countries and the strict denominational separation has a long tradition in Austria—alongside Catholic and Protestant religious education, Islam classes have been provided since 1982 and Orthodox classes since 1991 (Gmoser and Weirer 2019; Weirer 2013). Increasing religious diversity and non-denominationalism pose various organisational challenges, as in some schools, as many as eight denominational/religious communities offer separate classes simultaneously. Since 2022, all students in secondary schools who do not attend denominational religious education are required to take ethics classes instead.

The disconnect between theory and practice, as briefly outlined above, inspired us to embark on a research and development project under the title 'Christian–Islamic Religious Education in Team Teaching: Evidence-based development of local theories for didactics of collaborative religious teaching and learning processes' at the Faculty of Catholic Theological at the University of Graz in 2017.

The aim of the project 'Christian–Islamic Religious Education in Team Teaching' was to trial an exemplary model of collaborative Catholic and Islamic religious education within this specific legal and organisational framework over the course of several weeks. One integrative element was the collaborative creation of a specific teaching setting by an interreligious (Catholic–Islamic) project team. Muslim and Catholic students were taught by a team of Catholic and Islamic religion teachers. The collaborative religious education project was subsequently evaluated through multi-methodical analyses. The project is well documented and has produced five PhD dissertations (Gmoser 2023; Kramer 2023; Mešanović 2023; Wenig 2023; Yağdı 2023) as well as a joint anthology (Gmoser et al. 2024a).

The project set out to explore various aspects of interreligious learning processes in schools and to further develop didactics of collaborative religious teaching and learning processes. The different research strands, methodological approaches and perspectives were collated and combined in line with design-based research (Prediger and Link 2012; Gärtner 2018, 2022). As part of the project, the research team collected and incorporated data on students' learning preconditions as well as the background, habitual preconditions and competences of participating Islamic teachers, in particular. They also planned and carried out specific design experiments. The evaluation results gained thus formed the basis for local theories on interreligious learning processes (Gmoser et al. 2024a and chapter 7 of this article).

We consider interreligious education in school to be an integral part of religious education in a wider sense, as well as a specific religious pedagogical solution for the various challenges traditional religious education faces in the context of globalisation, religious pluralisation and the politicisation of religion and migration. At its core, this approach defines interreligious education as an intersubjective and equal exchange between members of different religions (Leimgruber 2012). The teachers serve as experts and institutional representatives of their respective religion and act as role models in how to communicate and talk about religion(s). The underpinning principle is that students of different religions learn together.

The present article first outlines the relevant legal framework for denomination-based religious education in Austria, as well as the conditions for cross-denominational collaborations. Then, we examine students' learning preconditions, with a focus on their preconceptions and attitudes towards religion in general, as well as Christianity and Islam in particular. Next, we analyse the working conditions of Islamic religion teachers at schools

in Austria as well as the required competencies for interreligious teaching processes. Important empirical insights extracted from lesson recordings, in-class observations and group discussions with several participating students are then collated and summarised. Finally, we present local theories based on these results and highlight further research potential.

## 2. Legal Framework

### 2.1. Interreligious Education Based on Denominational Frameworks

All forms of religious education are governed by the school legislation for denominational religious education (RE), as constitutionally enshrined in Article 17 [4] of the Basic Law on the General Rights of Nationals (StGG) of 1867 and guaranteed and defined in several legislative acts, first and foremost the Federal Law on the Religious Education of Children (RelUG) of 1949. According to this legal framework, responsibility for the provision of educational competencies and aims is shared by the state and church and religious communities. Curricula for all school types and grades, the training, selection and employment of religion teachers, the content and publication of textbooks and teaching resources, as well as the direct (content-related) supervision and oversight of subject auditors are within the purview of the relevant church and religious communities. The funding of teacher salaries, as well as organisational and disciplinary oversight, as carried out by school directors, are the responsibility of the state. Therefore, church and religious communities are granted a degree of autonomy, especially with regard to the provision of religious education, which consequently precludes students of one denomination from participating in religion classes of other denominations (cf. BMBWF 2023) and which gives them sole authority to regulate the provision (depending on the availability of religion teachers) and design of RE lessons (learning content and pedagogical competencies) (Kalb et al. 2003, p. 357; Kramer 2024, 127ff). Accordingly, all forms of interreligious education also lie within the exclusive purview of church and religious communities.

### 2.2. Interreligious Teaching—A Grey Area

Article 14 (5a) of the Federal Constitutional Law (B-VG) defines the elementary values of the school as "[d]emocracy, humanity, solidarity, peace and justice as well as openness and tolerance towards people [...] independent from origin, social situation and financial background [...]". The comprehensive development of children and teenagers should be enabled "[i]n a partnership" between students, parents and teachers to become "capable to take over responsibility for themselves, fellow human beings, environment and following generations, oriented in social, religious and moral values"; for example, to be able to "be open to political, religious and ideological thinking of others". § 2 (1) of the Austrian School Organisation Act (SchOG) contains almost identical wording. Upholding these elementary values and the willingness to foster a 'partnership' through open, tolerant and fair collaboration with people of different faiths and with different worldviews consequently opens up possibilities for interreligious education, although both legal texts only allude to such.

Cross-denominational or interreligious education is a direct collaboration between religion teachers of different religious communities in jointly designing and delivering lessons. Although such an approach is not explicitly mentioned in the legislation, neither is it precluded. Considering a literal as well as a teleological understanding of the legal texts and considering the religious pluralism of the Austrian student population, religious education should not and does not necessarily have to be the sole responsibility of a single church or religious community. Rather, creating a collaborative space for religious education facilitates interreligious dialogue between students of different faiths and thus enables them to learn about different beliefs, concepts, traditions, celebrations, etc. This can be achieved through interreligious team teaching and supported by a wider collaboration with students, parents/legal guardians, religion teachers, school boards, subject auditors and education authorities of participating church and religious communities. Our research project in Graz is an example of such an approach. This model of interreligious teaching complements RE lessons in providing a more comprehensive religious education that encourages openness

towards political, religious and ideological perspectives of peer students in line with the elementary values schools are required to deliver.

Although cross-denominational religious education is not yet legislated for by the government, several of the larger church and religious communities already implement interreligious models. The relevance and importance of such approaches is further evidenced in various laws that regulate the training of religion teachers: the Ordinance on Higher Education Curricula of 2013, the Teacher Education Act of 2005 and the Act on Quality Assurance in Higher Education of 2011, for example. Alongside the requirements for professional competencies, these legal texts specifically highlight the need for teachers to develop pedagogical skills (i.e., subject and didactic knowledge) as well as inclusive, intercultural, interreligious, social, diversity and gender competencies as part of their training. Many church and religious communities already include interreligious elements in their RE curricula, for example the Protestant and Catholic Churches, the Islamic Faith Community of Austria and the Alevi Religious Community of Austria.

### 2.3. Legal Considerations for the Implementation of Interreligious Education Models

In the absence of specific regulatory legislation, responsibility for a detailed framework of interreligious education, in particular with regard to planning, organisation and delivery, falls to the relevant church and religious communities. Joint interreligious lessons must be clearly defined as forming part of the respective denominational religious education. Therefore, we obtained express approval of the Catholic Church and the Islamic Faith Community for the temporary delivery of interreligious lessons and put various agreements in place before conducting our research project in Graz. These agreements were drawn up in collaboration with Catholic and Islamic teachers at different schools and outlined the following parameters: WHO (religion teachers), WHERE (schools), for WHOM (students), WHEN (dates and duration) and WHY (purpose and aims). WHAT (content), HOW (methods) and WITH WHAT (textbooks and teaching resources) were not specified so as to afford teachers maximum flexibility in designing lesson content and making appropriate decisions on teaching methods. Based on experience, additional measures to support the success of such projects are the early consultation of school boards/directors regarding organisational matters (e.g., scheduling parallel lessons) as well as engaging parents/legal guardians. Providing clear and comprehensive information about the lessons can help alleviate concerns about children being exposed to other religions and encourage participation. Although denominational RE lessons include interreligious elements, the principle of negative freedom of religion means that attending specific and temporary interreligious lessons remains optional. Additionally, full public transparency is necessary in order to proactively minimise political outcry and adverse media reporting.

## 3. Preconceptions and Attitudes as Learning Preconditions for Interreligious Education

Introducing interreligious education is often associated and met with high expectations (cf. our project design): The aim of bringing people of different faiths into dialogue is to facilitate mutual learning and to foster and strengthen openness towards religious plurality. The best outcome to hope and aim for is a reduction in religious biases, which also corresponds to schools' mandate to contribute towards a tolerant society. These goals and demands are of great importance and validity. However, they often overshadow the actual learning requirements of students, such as their specific preconceptions of other religions or associated views. On the other hand, numerous studies highlight the fact that religious prejudices and discrimination are not only present but on the increase in our society (Zick et al. 2011). Additionally, we know from developmental psychology that attitudes are already formed at a very early stage in childhood (York 2003, p. 3). It follows that interreligious education can never be taught under 'neutral' conditions. Students are already shaped by their caregivers and environment and thus bring their individual preconceptions and attitudes into the classroom. These learning preconditions, in turn,

affect their ability to process new information, their motivation and their learning success. Thus, it is essential to acknowledge and incorporate these conditions in didactic research. We specifically gathered insights into Christian and Muslim students' preconceptions and attitudes regarding religion as part of our project (Gmoser 2023) in order to reflect this.

### 3.1. Methodology

Information was gathered in four faith-specific group discussions with Christian and Muslim students, age 11–12. Every discussion was initiated with an open question: Students were asked to introduce themselves and explain to an extraterrestrial alien everything they need to know about religion(s) in order to understand this phenomenon. Based on the students' responses, the facilitator then raised further questions over the course of the discussion. The group discussions were analysed using the documentary method (Bohnsack 2014, pp. 33–69). This consists of a three-step process: Firstly, the entire material was subjected to what is known as 'formulating interpretation', before very dense passages were 'reflectively interpreted'. In the final 'comparative analysis', similarities and differences in the discourse or in the treatment of the same or a similar topic in different groups were identified.

### 3.2. Students' Preconceptions and Attitudes Towards Religion

Evaluating the collated data revealed different general approaches to religion amongst students: Catholic students demonstrated an analytical, impersonal approach to religion(s). They listed and categorised religions into major and minor categories without taking a personal position, completing the task from a more observational standpoint. In contrast, Muslim students positioned themselves as belonging to Islam from the outset of the discussion. Mentions of certain religious codes they observe everyday may indicate that religion has a stronger relevance in their daily lives.

All students—regardless of denomination—expressed a common positionality: They highly value freedom in religious contexts and firmly reject any coercion in or by religion. They made this clear, for example, when discussing whether infant baptism is compatible with a voluntary decision to belong to a religion or how the decision to wear the hijab is made.

During group discussions, students were further asked whether their friends' denomination was important to them. The intention was to ascertain how far their behaviour might be influenced by their preconceptions about religions. Bias research suggests that the degree to which a person's views determine their behaviour, for example, whether they choose to spend time or even establish friendships with members of a 'lower class', can be an indicator to measure someone's bias (Weiss and Hofmann 2016, p. 116). Although the students generally accepted religious plurality on a surface level (their friends' denomination holds no relevance for them), some findings suggested that they may still hold certain—implicit—prejudices about other religions. For instance, some students attached certain conditions to their acceptance of other religions (no infringement on freedom, and no pressure from people of different faiths). Other students mentioned conditions for converting to Islam in this context. Such references to a possible change of religion might indicate that they would prefer their friends to be Muslim, which in turn indicates that their openness towards other religions also has certain limitations.

### 3.3. Students' Preconceptions and Attitudes towards Christianity

As mentioned above, Catholic students spoke about Christianity from a more distant position. They described Christianity as the largest religious community in terms of numbers, but could not accurately explain the distinction between Christian denominations, for example. Discussions about religious biases revealed that Catholic students had never been challenged based on their faith and even when prompted by the facilitator were not able to name prejudices against Christians. Most likely, this was due to their social environment and their belonging to the religious majority in Austria.

Muslim students demonstrated a basic understanding of Christianity while also revealing clear gaps in knowledge. Interestingly, they brought a theological dimension to the discussion by identifying Jesus as the son of God. They were able to recognise that this concept distinguishes Christianity from their own religion, and some students expressed disagreement towards it. Overall, students assumed that the observance of religious rules was less important to Christians than to Muslims.

### 3.4. Students' Preconceptions and Attitudes towards Islam

Muslim students spoke with enthusiasm about their religion and welcomed questions by the (Christian) facilitator about Islam. Their answers demonstrated a level of confidence in subject knowledge as well as some gaps in knowledge. It was evident that there was a certain assumption of 'right' or 'wrong' in terms of how faith is expressed or experienced. In this context, 'right' was defined as coming from the heart, and this was seen as more important than outward performance.

Catholic students expressed an interest in Islam and demonstrated some basic knowledge. They also demonstrated a level of sensitivity in talking about Islam. For instance, one student expressed concerns about her questions being perceived as racist. However, discussions revealed that the Christian students also held socially widespread assumptions about Muslims, for example, that all Muslim women wear the hijab, that all Muslims have a refugee background, or that Islam is restrictive.

These insights highlighted the nuanced preconceptions students have towards religion(s), which need to be acknowledged as preconditions for their learning. Addressing religious diversity alongside specific religion-based preconceptions and attitudes in interreligious lessons creates space and opportunity to dismantle biases and thus contributes to achieving the core aims of interreligious education.

## 4. Conditions, Requirements and Interreligious Competencies for Religion Teachers

We live in a diverse society characterised by different cultures and religions, and this diversity is particularly reflected in schools. Teachers, therefore, play a crucial role in facilitating learning and understanding of different faiths. They put interreligious learning into practice and ensure that students develop democratic skills, thus supporting schools in fulfilling their essential educational aims. In turn, it becomes vitally important that teachers receive the necessary training to develop their own interreligious competencies. This undertaking raises various important questions: What makes interreligious teaching effective and inspiring? What are the skills and competencies required, and what support do teachers need in order to develop a respectful approach and enrich their appreciation of religious diversity?

The curriculum for Islamic religious education already includes interreligious learning (Güzel 2022, p. 203). Islamic religion teachers are called upon to implement these elements in order to help students develop interreligious competencies and to create space for exchange between teachers and students of different religions, cultures and worldviews in their lessons and the school environment.

A review of the literature on interreligious competencies revealed, however, that these are primarily directed at students and designed from a mainly Christian perspective (Mešanović 2023, p. 59). There is a clear theoretical gap with regard to specific interreligious competencies required in an Islamic context. Islamic religion teachers work under significantly different conditions and frameworks than the majority of their colleagues, which inevitably affects their interreligious competencies and how they can acquire and develop them. Therefore, we conducted a study as part of our research project (Mešanović 2023) that analysed the characteristics and effect of these conditions on Islamic religion teachers.

### 4.1. Methodology

The data were collected taking into account subjective structures of meaning (Hug and Poscheschnik 2010, p. 111) by using episodic interviews as a method for investigating the

working conditions and interreligious competencies of Islamic religion teachers. Qualitative research enables a description of lived experiences from the perspective of those affected, which allows for a deeper understanding of social realities and can make fundamental patterns and structures visible (Flick 2012, pp. 13–31). The focus here was on Islamic religious education teachers because they occupy a minority position in the Austrian context and their approach to interreligious learning was, therefore, of particular interest.

A code system was used to evaluate the results along two main dimensions: constitution and institution (Mešanović 2023, p. 366). The constitutional dimension concerns aspects that shape Islamic religion teachers' understanding of their role and determine the content and aims of Islamic religion lessons. This also includes the competencies and skills required for effective teaching and fulfilling their educational role. The institutional dimension dissects the frameworks and structures of the Islamic Faith Community of Austria (IGGÖ), the organisational structures in schools, as well as the interplay between different stakeholders (school board, colleagues and parents/legal guardians).

### 4.2. Constitutional Aspects

Aspects that affect the interreligious competencies of Islamic religion teachers include teacher image, the aims and status of religious education, the bidirectionality of interreligious learning, and professional skills training. The image of Islamic religion teachers is shaped by their self-perception as well as the external perception through school directors, colleagues and students, the status society awards teachers, and how they are represented in the media. Despite these pressures and challenges, teachers consider their work to hold important value and experience it as their lives' purpose. However, this self-awareness of responsibility in actively guiding young people in their development and acting as a role model can also become a source of stress.

Social constructs and preconceptions about religion teachers, particularly in the form of negative biases and stereotypes, as perpetuated by the media, further present limitations and challenges for teachers and create barriers for collaboration. Further, school boards and directors play a role in the development of religion teachers' interreligious competencies by way of the relationship they foster and their openness towards collaboration.

The teachers interviewed stressed the purpose of religious education and the important values of learning about and engaging with people from different backgrounds. They understand religious education as a safe space for students to express their feelings.

Interreligious exchange encourages dialogue and enriches students' personal development. However, it also holds potential for prejudices and conflict to emerge if not designed carefully. In this context, Islamic religion teachers highlighted the need for training opportunities at different stages in their careers to develop the necessary competencies. They asked for the provision of continued professional development and collaborative working experiences that promote interreligious skills as part of their early training.

### 4.3. Institutional Aspects

Islamic religion teachers pinpointed several challenges and working conditions that impact their daily work as a teacher and their organisational capabilities. These institutional aspects range from structural, organisational and human resources frameworks to cooperation between school boards, (subject) colleagues and parents/legal guardians.

The structural frameworks are established and implemented by the Islamic Faith Community of Austria (IGGÖ), as well as schools. The challenges arising from the IGGÖ framework include, first and foremost, unrealistic expectations in terms of teaching scope, compounded by a lack of material resources and adequate provision of interreligious training. Islamic religion teachers have to cover a large number of schools and deliver a high volume of lessons (up to 33 per week). This workload presents a barrier for integration with the schools' teaching staff (Mešanović and Weirer 2022, p. 264) and thus poses a hindrance to establishing collaborative relationships and initiating joint projects. The lack of teaching materials requires a refocussing of resources on lesson planning, thus

leaving little time for establishing teaching collaborations with colleagues. The frequent change of schools further hinders Islamic religion teachers in building these collaborative teaching relationships.

The institutional dimension further concerns frameworks governing human resources and interpersonal relationships with colleagues and staff. In this context, Islamic religion teachers reported experiences of exclusion due to prejudices and mistrust that they are faced with by the school leadership and other teachers. On the other hand, they stressed how important collaborative relationships between teachers are in order to facilitate interreligious exchange and design didactically sound lesson plans.

Teachers also considered it important to involve parents/legal guardians in this collaborative work. However, varying expectations and attitudes can be difficult to navigate.

The results of this study showed that working conditions significantly impact Islamic religion teachers in their teaching performance and personal/professional welfare, and hinder collaborations in educational institutions. Accordingly, it is evident that certain changes and improvements need to be made in order to support Islamic religion teachers in their professional development, reduce the pressures on teachers and promote better collaborative relationships.

## 5. Possibilities and Challenges of Interreligious Education in Team Teaching

As part of our research project, selected schools in Styria and Carinthia delivered collaborative Christian–Islamic lessons through team teaching. The interreligious elements had to be implemented within the framework of denominational RE, and this was achieved by temporarily teaching both religious groups in joint lessons. The respective religion teachers taught approximately three to five lessons together as a team. The research team observed these lessons in person (Kelle 2018) or recorded them on video (Fritzsche and Wagner-Willi 2015) for subsequent analysis and evaluation using the documentary method. The results gained from this first research cycle formed the basis for initial local teaching and learning theories (Gmoser et al. 2024b) regarding the possibilities and limitations of this teaching design. This process also yielded specific suggestions for improvement and next steps, which we implemented in a second research cycle that was carried out at an academic secondary school (Gymnasium) in Graz during the 2023/2024 academic year. The following presents insights gleaned from the first research cycle (Wenig 2024), in comparison with experiences and observations during the second research cycle, which implemented the suggested adaptations and changes. This comparative analysis provides a first glimpse of the effect and effectiveness of these improvements to the Christian–Islamic team-teaching model. However, the evaluation of the second cycle is still ongoing. Thus, we can only offer preliminary results and learnings that emerged from the different approaches between the first and second cycles at this time.

### 5.1. Interreligious Exchange in Christian–Islamic Team Teaching: Experiences and Insights

The collaborative religious teaching design through team teaching enables a mutual interreligious dialogue in that teachers present their respective faith, provide authentic information and tangibly demonstrate their religious practices. This allows students to focus on learning without the pressure to expertly represent their own religion. Instead, it offers them a space to speak about their personal religious background, express their experiences, reflect on their own position and grow their understanding in a safe learning environment guided by professional and pedagogically trained facilitators. The questions raised and topics discussed varied between each lesson. Students were highly engaged in this dialogue between themselves and their teachers in sharing their lived (religious) identities and worldviews. This was evident by how much attention they paid to the conversation and further demonstrated how important and welcome such exchanges between people of different faiths are in schools.

The results from this first analysis suggested that creating a dialogue through Christian–Islamic team teaching is both effective and valuable. However, the study also revealed the

complexity of didactically designing effective interreligious teaching and learning models in team settings. Their success is not guaranteed or incidental but depends significantly on the collaborative relationship between teachers and how they interact with each other and the students.

In the second cycle, the implementation of specific didactic measures during the preparation phase yielded clear improvements in terms of structure and communication in Christian–Islamic team teaching. The intention behind these measures was to acknowledge commonalities and differences between the teachers' religious, cultural and ideological viewpoints in order to identify and assess any potential for conflict or disagreement within the teaching team. This detailed, step-by-step approach to lesson planning not only applies to learning content, teaching methods and didactic tools, it also serves to clarify each teacher's role at every single phase in the teaching process. It aids in the decision-making as to who presents what information and when, and who takes the lead at different stages. Crucially, it also provides the space for both teachers to discuss their own relationship with their faith and their personal religious practices with each other. Such a structured approach establishes trust and promotes a supportive teaching environment, as both teachers know exactly what to expect, and what topics will be discussed and how, before going into the classroom. The critical analysis of different Christian and Islamic traditions can inspire a fruitful dialogue between students that encourages them to voice and reflect on their own views. Students as well as teachers are thus able to tangibly experience and learn that religiosity and spirituality, and how these are expressed, are subjective and vary from person to person. There is no right way to believe or practice a faith. In most forms of religion, thinking and behaviour are legitimate in their diversity. At the same time, extremist and radicalised forms of religion that are not conducive to life must be clearly named, critically questioned and rejected. The dialogue modelled between teachers is a key teaching tool that allows students to develop skills in speaking about religion(s). However, the success of any interreligious exchange and dialogue necessitates a base level of knowledge to build on. Therefore, interreligious teaching must first provide students with adequate information about their own and the other religion, respectively.

The results of our project showed that Christian–Islamic team teaching can deliver an effective and valuable model for interreligious education. It requires careful planning based on a clearly defined aim, the presentation of appropriate information, and must be approached with an open mind and respect. Teachers must examine and comprehensively familiarise themselves with the relevant subject matter both from a Christian and Islamic perspective as well as both in a theological and embodied dimension. This means going beyond the theoretical and understanding the diversity of religious practices within each religious tradition.

### 5.2. Challenges in Christian–Islamic Team Teaching

The first research cycle revealed that team teaching is more complex than initially assumed. Without a well-designed preparatory phase that ensures teachers are on the same page, a mutually independent approach can lead to competitive instead of collaborative behaviour. The specific challenges in interreligious team teaching are as follows: Shared lessons create an important space for interpersonal exchange but also generate binaries. The separation into Christian and Muslim learners effects an 'us and others' mindset that simultaneously highlights commonalities and delineates differences. Teachers thus face the task of providing introductory–level knowledge while also avoiding or countering stereotypes.

Teachers have to hold complex responsibilities and fulfil multiple functions: they are role models, facilitators and dialogue partners, all in one. Balancing the different demands, acting with gentle yet clear authority and creating a positive teacher–student relationship requires highly nuanced skills. They must be able to moderate discussions while also contributing their personal positions and experiences in a constructive manner. Navigating this duality between facilitating and participating in factual as well as emotional conversations requires a high degree of flexibility and competence, and the ability to

recognise when to switch roles. The teachers' tasks are to impart (theological) knowledge, ensure respectful and factually correct conversations, give equal space to students' and teachers' voices, and offer their own personal beliefs and views. The latter demonstrates that religion holds personal relevance, including for the teachers, and that they also bring their specific positionality to the classroom. However, this also blurs the lines between personal opinion and official dogmatics, as well as between traditional and modern attitudes. This coexistence of different perspectives adds a level of complexity and diversity to the teaching. Students and teachers can argue from equally valid perspectives and beliefs, which makes it more difficult to define or agree on clear and singular positions.

Thus, well-developed dialogue skills are an essential tool for religion teachers to deliver Christian–Islamic team teaching. An interreligious dialogue creates space for a nuanced and respectful discussion of differences: it lets differences exist without seeking to erase or homogenise them in favour of focussing on commonalities. Considering the high and complex demands of such a task, teachers must be afforded time and opportunity to continuously improve their dialogue skills in practice. It would be unrealistic to expect a dialogue-based teaching model to be perfectly implemented from the first lesson.

Our experience suggests that Christian–Islamic team teaching should be delivered in regular, temporary intervals that allow teachers to grow in their role and develop the manifold and necessary dialogue skills.

The varied challenges outlined above and the complex requirements to meet them demonstrate a clear need for interreligious team teaching to be integrated in the training curricula for religion teachers. Interreligious practice settings allow prospective teachers to develop the essential dialogue skills required for delivering effective interreligious education.

## 6. Students' Perspectives on Christian–Islamic Team Teaching

How students perceived and experienced the Christian–Islamic team teaching was analysed based on data gathered in post-lesson interviews with participating students. The following is an overview of the results gleaned from two such feedback sessions. The groups were specifically selected to represent a wide age range and equal balance between faiths. The first interview was conducted with year 4 students in a primary school after participating in three interreligious lessons. The shared lessons focussed on sacred buildings and the founders of religion(s). The second interview was conducted with year 10 students in an academic secondary school after they participated in five interreligious lessons. For this group, the shared lessons focussed on love and marriage in the Christian and Islamic traditions.

The interviews were structured as group discussions, beginning with an open impulse question to encourage students to speak as freely as possible about their experiences and impressions. The observations and insights gathered were evaluated and summarised into categories according to Mayring's content analysis (Mayring 2010). The following is an excerpt of the central findings that emerged.

### 6.1. Authentic Information

Students cited learning about religion (their own and the other) as a central component of these interreligious lessons. They appreciated the opportunity to speak to members of other faiths and, in particular, to discuss their respective views and beliefs in a mutual exchange. They experienced the lessons as a form of authentic knowledge sharing as opposed to media-based learning.

### 6.2. 'Us' and the 'Other': Religion as a Category of Difference

Contrary to our intention of not putting students in the role of didactic representatives, they automatically and almost exclusively spoke of 'their' and 'the other' religion. Evidently, the interreligious setting led to religion becoming a category of both identification and differentiation. However, this differentiation never took on a negative character. Rather,

it was an expression of curiosity and the desire to communicate their own religion as well as learn about the other religion.

### 6.3. Equal Exchange as the Basis, Prerequisite and Aim of Interreligious Learning and Teaching

Students (both Christian and Muslim) often raised equality as a central pillar for teaching and learning in various iterations. Interreligious learning is an equal and mutual exchange that must maintain its open and reciprocal principle. Students of both faiths need to be given equal space and time to express their views. It is the teachers' responsibility to ensure this is upheld. In this respect, students displayed a strong sense of awareness and criticised the didactic disparity in traditional denominational religious education. They expressly demanded a subject-oriented teaching model that adequately addresses their own (religious) experiences, knowledge and questions.

### 6.4. Finding Commonalities and Differences between Islam and Christianity

Students attached great importance to both discovering communalities and exploring differences through interreligious teaching and learning processes. It afforded them the opportunity to realise that they have more in common than they perhaps assumed. At the same time, they could explore their curiosity about the characteristic differences between religions. Openly discussing diverse views and values enabled them to find common ground without dismissing or negatively judging respective differences.

### 6.5. Curiosity about Religious, Ritual and Spiritual Practices and Places

Beyond conceptual characteristics, students also expressed great interest in learning about ritual, religious and worship practices as well as associated sacred places of the other religion. This demonstrates a clear need for incorporating performative dimensions in interreligious education. Younger students were particularly curious about traditional clothing as a visible marker of religious practice and roles, which can be an important access point for developing empathy.

### 6.6. Interreligious Learning to Dismantle Biases and Generalisations

Students praised the opportunity to address and deconstruct biases and stereotypes about the 'other' religion as a particular benefit of interreligious lessons. One of the tenets of interreligious education is to deconstruct generalisations and promote fact-checking. Subject knowledge and research open up a more nuanced perspective on topics that are often presented through a narrow, generalised and Islam-focussed lens in the media.

### 6.7. Openness and Respect as the Basis for Dialogue and Interreligious Education

Students expressed in various ways that open communication and a respectful environment are key prerequisites for interreligious teaching and learning processes to be successful. In contrast to other settings, schools are seen as a place to explore commonalities and differences between religions in a safe and open manner. Thus, it is essential that teachers create an atmosphere of trust and encourage mutual listening and understanding. This enables students to experience and practice a pluralist approach towards diversity in religion.

The results of these interviews yielded important impulses for further empirical research on interreligious teaching and learning processes. Religious pedagogy and educational models should reflect the diversity of our increasingly pluralistic society in comprehensive didactic approaches. In order for religious education to maintain relevance in students' lives and to support them in developing their own religious identity, it must take their experiences and needs into account.

## 7. Integrating Local Theories in Collaborative Religious Didactics for Teaching and Learning Processes

Presented above are various insights gained from our project 'Christian–Islamic Religious Education in Team Teaching'. In accordance with our design-based research approach, the following conclusions extrapolate local teaching and learning theories in support of didactics for collaborative religious education. In contrast to 'big' theories, 'local' theories focus on representativeness across subject level, while also incorporating case-specific observations that indicate wider applicable requirements and conditions for effective teaching and learning processes.

### 7.1. Interreligious Education Faces High Expectations from Religious Pedagogy, Educational Policymakers and Schools. These Pressures Add an Emotional Dimension to the Teaching Model and Bear the Risk of Causing Teachers to Feel Overwhelmed. Whether or Not These Expectations Can Be Fulfilled in Schools Depends on a Broad Range of Circumstances and Conditions

Since interreligious teaching and learning processes entered religious pedagogical discussion, they have been ascribed various intentions. Students should acquire basic knowledge about their own as well as other religions, they should be able to identify commonalities and differences between the respective religions, they should learn to interact with people of different faiths in a respectful and appreciative manner, they should practice taking different perspectives, and they should affirm religion(s)' claim to truth while also recognising aspects of truth in both their own as well as other religions, despite relativist tendencies, which enables them to adopt a pluralist attitude towards religious diversity. At the same time, students should also develop more awareness of the diversity contained within each religion (Unser 2022, 2018; Meyer 2019).

School boards and education policymakers intend interreligious education as a solution for reducing (religion-based) conflict in schools, preventing religious radicalisation, and promoting peaceful coexistence in a democratic society. Such multi-layered and high-aiming expectations are difficult to realise in practice. The success of interreligious education is determined by the various structural and individual conditions that affect the teaching environment (e.g., linking lessons between different church and religious communities, afternoon teaching slots, and Islamic religion teachers having a high teaching load and working in many different schools). A crucial prerequisite for the delivery of interreligious learning is that religion teachers possess the necessary competencies and demonstrate an actively collaborative approach.

### 7.2. Interreligious School Projects Must Take the Social Environment, Political Climate and News Reporting/Media IMAGES into Account

Demographic changes and growing religious/cultural plurality also lead to shifts in the relationship between majority and minority groups. The political instrumentalization of different social groups introduces a rift between people that can further generate conflict. These pressures and challenges are particularly heightened in schools as a concentrated shared space for learning and development. For team teaching to be effective and to enable inclusive and equal conversations, teachers must be aware of the different needs and concerns of the majority as well as minority student populations. Identifying these and their potential negative impacts in the classroom allows teachers to put measures in place to avoid potential conflict from the outset.

### 7.3. Collaborative Religious Team Teaching Requires an Intensive Preparation and Planning Period, Where Both Teachers Explore Theological, Religious Pedagogical and Didactic Concepts, Assumptions and Aims Together. Topics Related to Students' Lived and Daily Experiences Form the Centre of Religious Discussion, in Accordance with Subject-Oriented Teaching Models and to Leverage the Particular Teaching Setting

We recommend that participating teachers incorporate trust-building measures in their joint lesson planning to establish a supportive and fruitful working relationship. Good communication and joint preparation provide the fundamental basis for critically and constructively reflecting and exploring religion(s) from different perspectives. This

enables teachers to better understand each other's viewpoints and to agree on shared principles and values that form the pillars of their professional collaboration. It also serves to pre-empt parallel, unrelated or confrontational behaviour in the classroom. In a further step, teachers must jointly define clear religious–didactic premises and aims for their collaborative lesson design. They must consider both advantages and disadvantages of the didactic setting (e.g., the balance between teacher-centric and student-centric approaches) and make necessary adaptations, in line with the overall teaching aim.

### 7.4. Successful Collaborative Interreligious Team TEACHING—Preferably in the Form of 'Co-Construction'—Requires Teachers to Actively Reflect on Their Respective as Well as Shared Teaching Roles (Together)

Interreligious team teaching should form a collaboration in the sense of 'co-construction'. This means that teachers continuously share and link (co-construct) their knowledge, thus expanding it, and then develop a common approach in line with their defined goals (Gräsel et al. 2006, pp. 210–11). Constructive team teaching, thus, is a constantly evolving process. Clarifying shared and individual roles and responsibilities ensures a fair balance and pre-empts any potential for conflict or competitive attitudes. A particular and particularly important aspect of collaborative religious team teaching is that teachers take on the position of role models. Christian and Islamic teachers lead by example in how they interact and speak with each other, including non-verbal signs of engagement and respect, thus demonstrating to students how members of different religions can be in equal dialogue with each other and discuss potentially diverging worldviews without judgment or missionary intent. Teachers must be aware of their responsibilities in this function.

### 7.5. Students Hold Various Roles in Interreligious Teaching Models. While Some Students Step into Certain Roles with Willingness and Confidence, or Adopt Them Automatically, Some Might Feel Pushed into These Roles or That They Are Being Assigned Them by Their Teachers and Peers. It Is Paramount That Interreligious Lessons Are Safe Spaces Where Students Are Allowed to and Able to Determine Their Own Roles

The analysis of observations and recordings showed that some students, Muslim students in particular, often stepped into the role of experts about their respective religion and were keen to share their knowledge and experiences, as well as their beliefs and traditions, with others, thus bringing their personal religiosity to the discussion. However, we cannot assume that all students assume these representative roles of their own accord or comfortably. Therefore, teachers need to exercise sensitivity and awareness in order to ensure students are not pushed into the position of expert about 'their' religion in interreligious learning and teaching processes. The analysed data further revealed the likelihood for students to feel overwhelmed if forced into a role they do not feel comfortable in, for instance, if they are expected to authentically represent or explain religious teachings, rites, values or practices. Teachers must actively pre-empt situations where students have to take on a representative role and speak on behalf of their respective religious community/institution. Instead, they need to collaboratively create a safe interreligious space that allows students of all faiths to share their personal beliefs openly and freely with others and engage with students from different religious communities in a direct dialogue about their different views. Students should be enabled to freely express and position themselves. These skills in recognising, voicing and discussing differences reach far beyond religious contexts and will serve them as invaluable tools throughout all aspects of their lives.

### 7.6. Interreligious Teaching Models Need to Take Students' Preconceptions and Attitudes about Religion(s) as Well as Specific Prejudices into Account and Address Them

In accordance with research on learning processes, students' learning preconditions significantly impact their learning and learning success. In the context of interreligious education, these preconditions also include biases towards religion and other religions. Thus, it will be beneficial for teachers to ascertain their students' attitudes towards religious

diversity and (other) religion(s) prior to or at the outset of interreligious lessons in order to make necessary didactic adjustments. Further value lies in addressing specific biases held by students or prejudices prevalent in society at large, as this allows teachers and students to dismantle them through interreligious processes. This task requires teachers to first acknowledge their own (often implicit) religion-based biases and attitudes in order to ensure that they do not inadvertently introduce their personal principles and communication patterns into the classroom, dismiss diverging perspectives or (mis)judge students' approaches and contributions.

*7.7. Explicitly Discussing Religious Differences between Teachers and Students in Christian–Islamic Team Teaching Can Give Rise to Religious Othering, an Inherent Potential That Cannot Be Completely Eliminated. Thus, One Aim of Interreligious Education Is to Instil Awareness and Sensibility for Othering Phenomena and Their Negative Effects in Both Teachers and Students*

'Religious othering' is defined as constructing an 'other' that is devalued in favour of a dominant 'own'. In this context, it is important to consider the consequences and emotional impact labels such as 'other' or 'foreign' have for the people/group that are thus spoken about. For Joachim Willems, this is related to "religionist racism" (Willems 2021, p. 478), a concept that acknowledges the potential intersection between differentiating categories of religion, culture and ethnicity, and language, nationality and gender. Islamophobia—like other forms of group-based devaluation, such as Antisemitism—is affected by these overlapping categorisations. (Inter)religious education processes must take these realities into account and investigate the language used in teaching settings. On the one hand, it can—intentionally or unintentionally—reinforce boundaries of difference and thus reproduce dominance (Mecheril and Melter 2010, p. 176). On the other hand, it can create space for critically addressing and deconstructing structures of exclusion and categorisation that are at play in non-religious and non-theological (e.g., political) discourses (Willems 2021, p. 484) in order to instil sensibility for 'religionist' racism. Understanding different forms of racism (Willems 2021, p. 482) then enables people to recognise mechanisms of external categorisation for the purpose of devaluation and mechanism of othering. This provides the foundation for developing anti-racist teaching and learning processes. This process also requires a self-critical reflection of religious—and in particular, interreligious—teaching practices as to whether this exchange between different religions also gives rise to stereotypes and derogatory images by associating certain characteristics, markers and behaviours with certain religions (Willems 2022, p. 246). One way to introduce the concept of diversity is to highlight and explicitly discuss the different interpretations of scripture, the various religious practices and the multiple ethical principles that (co)exist within the same religious community.

*7.8. Interreligious Teaching and Learning about Commonalities and Differences between Different Religious Worldviews Is Most Effective and Enriching When Teachers Can Present as Many Nuanced Perspectives as Possible, without Ignoring Differences. On the Contrary, These Can Be Harnessed as Opportunities to Dismantle and Complexify Binary Arguments*

The lessons observed during this research project predominantly focussed on commonalities between Islam and Christianity, while avoiding discussions about differences. If the issue was raised, for example, in student questions, teachers quickly redirected the conversation. In order to fully leverage the potential of this particular didactic setting, teachers should explore as many different and nuanced perspectives on relevant topics as possible as part of their lesson preparation and present them without shying away from differences. Learning about nuances in interpretation and discussing religion(s) from individual viewpoints enables students to overcome binary patterns of thinking (in terms of 'allowed' or 'forbidden', 'right' or 'wrong'). Deconstructing black-and-white arguments in this manner encourages students to develop their own positionality beyond binary constructs and thus plays a central function in religious education for preventing fundamentalist attitudes (Weirer 2021).

The above-presented theories also highlight elements for further consideration and investigation in subsequent research cycles on Christian–Islamic team teaching, in line with design-based research. Yet, they also already offer inspiration and a foundation for collaborative religious teaching and learning didactics that equally apply to other forms of collaborative religious education models.

**Author Contributions:** All authors have collectively contributed to the conceptualization, methodology, writing, and editing of this manuscript. They have collaboratively developed and finalized all aspects of the work. All authors have read and agreed to the published version of the manuscript.

**Funding:** This research was funded by the Austrian Science Fund (FWF; grant DOI: P34282). For the purpose of open access, the authors have applied a CC BY public copyright licence to any Author Accepted Manuscript version arising from this submission.

**Institutional Review Board Statement:** A statement from the Ethics Committee of the University of Graz was not necessary, as the ethical aspects of the study involving humans had already been examined in advance as part of the review by the Austrian Science Fund (FWF). The two experts explicitly confirmed that the concept of the study fulfils all ethical standards.

**Informed Consent Statement:** Informed consent was obtained from all subjects involved in the study.

**Data Availability Statement:** The raw data supporting the conclusions of this article will be made available by the authors upon request.

**Conflicts of Interest:** The authors declare no conflict of interest. The funders had no role in the design of the study; in the collection, analyses, or interpretation of data; in the writing of the manuscript, or in the decision to publish the results.

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
