# Peer review of "Shared Religious Education through Christian–Islamic Team Teaching"

_religions, doi:10.3390/rel15091068_

Round 1

Reviewer 1 Report

Comments and Suggestions for Authors

Explanations and clarity:

211 – I would have found a phrase explaining the documentary-method helpful

224-5 – It is not clear what the views on baptism and hijab may be

302-308 – ‘qualitative method’ and ‘episodic interviews’. It is not clear to me exactly what was done when by whom. A little more explanation would help.

Section 4 – in this whole section it’s not clear to me why Muslim teachers were singled out. Was there no exploration of Christian teachers? If not, why not?

433-4 – Are there really no boundaries? These statements are extreme. In line 597 you seem to want to counter radicalization, in which case you must have some limits in mind.

703-4 – It is not just Islamophobia that is an issue. Different communities exhibit and suffer from different prejudices including Antisemitism, Christianophobia etc

731 “without shying away from difference” is hugely important. This exercise should never ignore difficult questions or topics. Young people are just as aware of them as adults. In their preparation co-teachers should be careful not to plan to ignore or sanitise difference and difficult topics but rather address them sensitively – as I believe your report is indicating.

There are just a few places where the otherwise excellent English needs some attention or typos correcting:

45 can choose to opt in

40 past tense after present perfect in 39 sounds strange

76 the evaluation results gained thus – would sound better

95 delete first

217 major and minor

227 ascertain how far (delete in)

392 glimpse of (not at)

410 this this

419 communalities – commonalities?

525 pillar

560 a place to explore

Comments on the Quality of English Language

Fine with corrections above although maybe get it checked through once again

Author Response

Thank you very much for taking the time to review this manuscript and for your valuable and helpful feedback. Below you will find some responses to your comments. The relevant revisions/corrections have been highlighted in yellow in the newly submitted documents.

Point-by-point response to Comments and Suggestions for Authors

Comments 1: I would have found a phrase explaining the documentary-method helpful.

Response 1: We hope that this brief insight into the documentary method according to Bohnsack has helped to clarify matters. Unfortunately, a more detailed explanation is not possible due to the number of characters.

Comments 2: It is not clear what the views on baptism and hijab may be.

Response 2: The sentence “This is expressed in the group discussions, for example, in conversations about the importance of freedom in religious decisions in connection with infant baptism or the wearing of the headscarf” was changed to “They made this clear, for example, when discussing whether infant baptism is compatible with a voluntary decision to belong to a religion or how the decision to wear the hijab is made.” This will hopefully make it clearer what the children were talking about.

Comments 3: ‘qualitative method’ and ‘episodic interviews’. It is not clear to me exactly what was done when by whom. A little more explanation would help.

Response 3: Thank you for pointing out that this wording can lead to confusion about the specific methodology. We have changed the explanation of the methodology in the article so that it is now clearer that episodic interviews (i.e. a qualitative method) were used.

Comments 4: in this whole section it’s not clear to me why Muslim teachers were singled out. Was there no exploration of Christian teachers? If not, why not?

Response 4: To answer this, we have inserted the following sentence: “The focus here was on Islamic religious education teachers because they occupy a minority position in the Austrian context and their approach to interreligious learning was therefore of particular interest.”

Comments 5: Are there really no boundaries? These statements are extreme. In line 597 you seem to want to counter radicalization, in which case you must have some limits in mind.

Response 5: Thank you for this important comment. We have added the following sentence for clarification: “At the same time, extremist and radicalised forms of religion that are not conducive to life must be clearly named, critically questioned and rejected.”

Comments 6: “without shying away from difference” is hugely important. This exercise should never ignore difficult questions or topics. Young people are just as aware of them as adults. In their preparation co-teachers should be careful not to plan to ignore or sanitise difference and difficult topics but rather address them sensitively – as I believe your report is indicating.

Response 6: We fully agree with this view and see this fact as an important result of our project.

3. Response to Comments on the Quality of English Language

Point 1: There are just a few places where the otherwise excellent English needs some attention or typos correcting.

Response 1: Thank you very much for the close look at the English language and the comments on some of the wording and typos.

Reviewer 2 Report

Comments and Suggestions for Authors

Review of “Christian-Islamic Religious Education in Team Teaching” Religions

Key Frames:

Page 1:

Three issues: interreligious celebrations, projects, and collaborative teaching

Cross-denominational and interreligious collaborative teaching models for several years

Fundamental academic reflection on subject - excellent as a starting point, could build beyond to consider other European models, etc. 

Those responsible are those who are “officially recognized" church and religious communities (16 total with 15 offering denominational religious education). It might be useful to consider those outside the "officially recognized" organizations. Is this simply done by acculturation into a dominant (i.e. officially recognized) denomination for either Christian or Islamic pupils? 

Long-standing legal frameworks are now being stressed because of “increasing religious diversity and non-denominationalism". This point needs much more data and evaluation of that data to really be helpful here. It is one of the most interesting points made in the early paper, but it could be strengthened and might even be an article unto itself. 

 Comments: 

This article does a very good job of outlining some of the reasons for the pre-conditions of the Austrian context and why such efforts are being attempted in the space of Christian-Islamic Religious Education in Team Teaching settings. The article has a clear objective, to present the purposes and challenges of facilitating and conducting interreligious education in school settings and also to provide best-practices learned along the way.

The articles specificity is productive here because it allows for a more focused look at one set of circumstances rather than trying to define the entirety of the religious education needs in Austria. The author(s) have expertly placed the complexity of the situation of front but not let the complexity overshadow some of the real achievements they seem to be making along the way. The author(s) claim that this project produced five PhD dissertations and a joint anthology. One might wonder where this article can situate itself in the context of so much other produced literature related directly to this one project. The legal framework that they are describing in the first section is useful but unique, so thinking through what comparable models exist and if the same successes and challenges exist would be useful information and also help see where the real innovation of this particular project lies. It does seem that although cross-denominational religious education is not yet required by law, that the claimed success of this project is likely seeking to influence that purpose.

The most useful section is the consecutive 3.2 – 3.4 where they show some of the general data about how students perceive religion, Christianity, and Islam. The results are not unexpected, but good on the authors for including this information in the article as it lays out the seedbed within which this project bears fruit. As well the competencies section was very interesting for religion teachers (section 4). The assumption coming in to the article was that students would prefigure heavily in this, but the specific focus on the teaching aspects proved useful. The negotiation of team teaching pairs makes this article potentially important to the broad concept of team teaching and pedagogical strategy, so it can speak to audiences in general ways beyond the interreligious context.

Some general advice/considerations on the article:

It might be interesting to look at the definition of interreligious teaching (around lines 133-5) and see if this is overly specific and leaving out some spaces were this term might apply in other ways. More specifically, are there legal definitions that could be brought into the article to help understand what the aims of the legislation are, what problem they seek to prevent or solve, and how the government will evaluate success in this realm.

Author Response

Thank you very much for taking the time to review this manuscript and for your valuable and helpful feedback. Below you will find some responses to your comments. The relevant revisions/corrections have been highlighted in yellow in the newly submitted documents.

Point-by-point response to Comments and Suggestions for Authors

Comments 1: Those responsible are those who are “officially recognized" church and religious communities (16 total with 15 offering denominational religious education). It might be useful to consider those outside the "officially recognized" organizations. Is this simply done by acculturation into a dominant (i.e. officially recognized) denomination for either Christian or Islamic pupils?

Response 1: In Austrian school law, pupils who do not belong to an officially recognised religion are categorised as "without confession". These pupils do not have denominational religious education, but (at secondary level 2) ethics lessons.

Comments 2: Long-standing legal frameworks are now being stressed because of “increasing religious diversity and non-denominationalism". This point needs much more data and evaluation of that data to really be helpful here. It is one of the most interesting points made in the early paper, but it could be strengthened and might even be an article unto itself.

Response 2: Unfortunately, there is no valid data on the pupils in the individual religious education programmes that is collected across all religions and denominations. We agree that this data would be highly relevant.

Comments 3: The author(s) claim that this project produced five PhD dissertations and a joint anthology. One might wonder where this article can situate itself in the context of so much other produced literature related directly to this one project.

Response 3: This article attempts to summarise the most important results from the five dissertations from the perspective of interreligious didactics.

Comments 4: The legal framework that they are describing in the first section is useful but unique, so thinking through what comparable models exist and if the same successes and challenges exist would be useful information and also help see where the real innovation of this particular project lies. It does seem that although cross-denominational religious education is not yet required by law, that the claimed success of this project is likely seeking to influence that purpose.

Response 4: We understand that this impression arises. Nevertheless, we do not want to fundamentally change the legal form of organisation of religious education, but rather encourage cooperation between different religious schools and support this through appropriate didactics. This requires fundamental research in this area, which we wanted to provide with this project.

Comments 5: It might be interesting to look at the definition of interreligious teaching (around lines 133-5) and see if this is overly specific and leaving out some spaces were this term might apply in other ways. More specifically, are there legal definitions that could be brought into the article to help understand what the aims of the legislation are, what problem they seek to prevent or solve, and how the government will evaluate success in this realm.

Response 5: The legal basis for denominational religious education in Austria dates back to 1949 and has only been adapted slightly since then. Interreligious education is not recognised in Austrian school legislation. Therefore, projects such as the one described are an attempt to try out co-operation between different denominations without leaving the legal framework.